# For Better or for Worse? A Scoping Review of the Relationship between Internet Use and Mental Health in Older Adults during the COVID-19 Pandemic

**DOI:** 10.3390/ijerph19063658

**Published:** 2022-03-19

**Authors:** Hui Foh Foong, Sook Yee Lim, Fakhrul Zaman Rokhani, Mohamad Fazdillah Bagat, Siti Farra Zillah Abdullah, Tengku Aizan Hamid, Siti Anom Ahmad

**Affiliations:** 1Malaysian Research Institute on Ageing (MyAgeingTM), Universiti Putra Malaysia, Serdang 43400, Malaysia; huifoh@upm.edu.my (H.F.F.); fzr@upm.edu.my (F.Z.R.); fadilbagat@upm.edu.my (M.F.B.); farra@upm.edu.my (S.F.Z.A.); 2Faculty of Applied Sciences, UCSI University, Cheras, Wilayah Persekutuan Kuala Lumpur 56000, Malaysia; limsy@ucsiuniversity.edu.my; 3Department of Computer and Communication Systems Engineering, Faculty of Engineering, Universiti Putra Malaysia, Serdang 43400, Malaysia; 4Department of Electrical and Electronic Engineering, Faculty of Engineering, Universiti Putra Malaysia, Serdang 43400, Malaysia

**Keywords:** COVID-19, Internet use, mental health, older adults, social media

## Abstract

Older adults were advised to avoid social activities during the outbreak of COVID-19. Consequently, they no longer received the social and emotional support they had gained from such activities. Internet use might be a solution to remedy the situation. Therefore, this scoping review sought to map the literature on Internet use and mental health in the older population during the pandemic to examine the extent and nature of the research. A scoping review was conducted using eight databases—PubMed, Scopus, Ebscohost Medline, Ebscohost Academic Search, Ebscohost CINAHL Plus, Ebscohost Cochrane, Ebscohost Psychology and Behavioural Sciences Collection, and Ebscohost SPORTDiscus, according to PRISMA guidelines. Two pre-tested templates (quantitative and qualitative studies) were developed to extract data and perform descriptive analysis and thematic summary. A total of ten articles met the eligibility criteria. Seven out of ten studies were quantitative, while the remainder were qualitative. Five common themes were identified from all the included studies. Our review revealed that Internet use for communication purposes seems to be associated with better mental health in older adults during the COVID-19 pandemic. Directions for future research and limitations of review are also discussed.

## 1. Introduction

### 1.1. COVID-19 Pandemic and Older Adults 

Coronavirus Disease 2019 (COVID-19) has impacted the world since December 2019. Due to the absence of effective medicines and the varied vaccination rate across different countries during the initial occurrence of the pandemic, non-pharmacological interventions such as social distancing, wearing a face mask, frequent handwashing, and home quarantine were deemed the only solutions to mitigate the spread of the COVID-19 virus. Although non-pharmacology interventions such as social distancing and home quarantine have effectively reduced the risk of COVID-19 infection, the interventions also introduced negative psychological impacts to society [1].

Older adults were the most affected and vulnerable group during the outbreak of the COVID-19 pandemic as they have been, and continue to be, considered at risk for severe complications and mortality from COVID-19 infection [2]. Due to their vulnerability, older adults were advised to stay at home and avoid social activities as well as non-essential social interactions. However, due to these restrictions, the older people no longer received the social and emotional support that they had previously gained from social activities and interactions. Consequently, older people are at a higher risk of developing mental health issues such as loneliness, social isolation, and depression throughout the pandemic period [3,4]. According to Polenick and colleagues, in a sample of 701 older adults aged 50 and above living with morbidity from the United States, more than half (66.4%) of them had moderate to severe levels of loneliness. Other COVID-19 related psychological disturbances reported in this study were anxiety symptoms, excessive worrying about COVID-19 infection, and financial difficulties [5]. According to a study in Switzerland, the prevalence of loneliness in older people increased after the implementation of physical distancing and factors associated with loneliness in older adults were being female, poverty, living alone, individuals with no children, individuals unsatisfied with their contact with neighbours, and individuals interviewed after the physical distancing recommendations [6]. Furthermore, in London, a total of 12.8% of older adults reported feeling worse on the depression symptoms, and 12.3% reported feeling worse on the anxiety symptoms during the national lockdown period. Factors associated with high depressive and anxiety symptoms were being female, younger age, non-married, disturbed sleep pattern, feelings of loneliness, and living alone [7].

### 1.2. Internet Use and Mental Health in Older Adults

Internet use became a trend during and after the COVID-19 pandemic, as most services have now gone online. In other words, people can now enjoy the convenience of grocery shopping, managing their finance, paying bills, working from home, and casually interacting with others online without leaving the house. Internet usage has risen from 40% to 100% compared to the pre-lockdown period [8]. In addition, video-conferencing services had a ten times increase in use, and delivery services had a 30% increase [8]. However, older adults are not in the age group that could reap the maximum benefits from Internet use as they are deemed not technology savvy and lack digital literacy [9,10]. The Internet, together with electronic gadgets, could be the solution to continue social interaction and support for older people during the pandemic. Most social technologies are now freely available, allowing people to virtually connect despite the social restrictions posted due to the pandemic. 

The previous literature on the relationship between Internet use and mental health in older persons were inconclusive. For example, in England, Internet use for communication purposes was associated with lower depressive symptoms and higher life satisfaction, whereas Internet use for information seeking was associated with lower life satisfaction [11]. In contrast, a longitudinal study in China concluded that Internet use was associated with the increased incidence of depressive symptoms in older adults and the negative impacts on mental health were more evident in women, the young and middle-aged, those with high-income, less educated, and living with others [12]. The different findings could be attributed to the different purposes of Internet use adopted by older persons. 

### 1.3. The Need for This Scoping Review

This scoping review identified quantitative and qualitative studies of the related study topic. The included studies were aggregated and evaluated to produce an accurate response to the research question that needed to be answered [13]. To the best of our knowledge, there has been a systematic review that examined the relationship between Internet use and mental health in older adults [14], but there have been no reviews on this topic during the pandemic period. Due to the restrictions during the pandemic, most of the older adults could not interact with others outside their home, thus the usage of the Internet could be a possible solution to remedy the situation. Therefore, a review on the experience of using the Internet during the pandemic on their mental health is essential. The present review attempts to map the literature on Internet use and mental health in older people to evaluate the extent and nature of these research activities by using the available literature from both quantitative and qualitative studies.

## 2. Methodology

### 2.1. Research Question

This scoping review was developed by following the guidelines from Arksey and O’Malley, which starts with formulating a research question [15]. A research question was developed based on the population, intervention, controls, and outcomes [16]. The research question of this review was “What is the extent and nature of research activities on Internet use and mental health in older adults during the COVID-19 pandemic?”

### 2.2. Search Strategy

Table 1 shows the keywords used in the search strategy. Eight databases were searched independently by two investigators on 12 October 2021 for potential studies. The databases involved were PubMed, Scopus, Ebscohost Medline, Ebscohost Academic Search, Ebscohost CINAHL Plus, Ebscohost Cochrane, Ebscohost Psychology and Behavioural Sciences Collection, and Ebscohost SPORTDiscus. We limited the years of publication to 2019–October 2021 as we were only interested in including studies that were conducted during the COVID-19 pandemic. 

### 2.3. Eligibility Criteria 

Table 2 presents the rationales of the inclusion and exclusion criteria of this scoping review. The inclusion and exclusion criteria were: (a)Studies that were conducted during the COVID-19 pandemic;(b)Studies from any country and any population;(c)Studies that were only published in the English language. There was no research design limitation. However, publications without research elements such as commentary, letters for editor, and study protocol were not included;(d)Studies that applied quantitative statistical analyses as well as used qualitative data methodologies;(e)Studies that must contain both Internet use and mental health variables. Studies that were missing either one element were excluded. The Internet is a vast network that connects technology devices all over the world, Internet use referred to Internet access points to personal or public computers, or connected devices such as smart phones and tablets, or technology applications. Examples of mental health variables were, but not only limited to, loneliness, depressive symptoms, social isolation, suicidal ideation, life satisfaction, and flourishing; and(f)Studies involving older adults only or older adults as one of the sub-groups of research sample/respondents.

### 2.4. Screening

Potential studies from the eight databases were imported into the EndNote Program X5 by two independent authors. Then, duplicate articles were removed by the Endnote Program X5 and another round of manual checking of duplicate articles was performed by the two independent authors. Next, the two authors screened titles and abstracts for feasibility based on the inclusion and exclusion criteria set beforehand. Furthermore, the authors also performed a manual hand-search for more potential studies to be included from the citation in the included articles. Both authors agreed with all ten inclusive papers after discussion with a third author. Please refer to Figure 1 for the review flowchart.

### 2.5. Data Extraction

We developed two templates to populate data from the included studies. One table was for the quantitative studies, whereas another template was for qualitative studies. We found that populating the studies’ information based on quantitative and qualitative designs was more feasible and systematic after piloting the table. The format of the templates was designed by including the possible information that we needed for the thematic synthesis. After developing the templates, authors filled in the templates during the data extraction and found that the templates were feasible to be used. The quantitative template consisted of the authors’ names and year of publication, study aim, country or setting, research design, sample size, sample characteristics, Internet use measure, mental health measure, and principal findings.

### 2.6. Data Analysis 

The authors first read, understood, and synthesised the findings of the included studies. Then, group discussions among the authors were conducted to discuss the analytical strategy. The authors analysed the results by categorising them into different common themes [17]. The themes generated in this review were (i) individual differences in mental health during the COVID-19 pandemic; (ii) patterns of Internet use; (iii) the relationship between Internet use and mental health: comparison by different age groups; (iv) Internet use and mental health; and (v) difficulties in using ICTs. 

## 3. Results

### 3.1. Study Characteristics

All the quantitative and qualitative studies are summarised in Table 3 and Table 4, respectively. A total of ten studies were included in this scoping review. Out of the ten studies, there were seven quantitative studies and three qualitative studies. For the quantitative studies, most of them were cross-sectional studies (*n* = 6), while only one was an intervention study (*n* = 1). Out of the six cross-sectional studies, only two involved secondary data analyses, whereas the rest were primary data analysis. Most of the studies were contributed from European countries (*n* = 9), and only one study was contributed from Asia—Hong Kong. For sample size in the quantitative studies, the range of sample size was from 101 to 3810. Three studies had a sample size of less than 500, whereas the rest had a sample size of more than 1000. There were only two quantitative studies that involved only older adults as the sample, whereas the rest involved multiple age groups. Out of the five studies that involved multiple age groups, three studies performed a results comparison among multiple age groups. 

There were three qualitative studies included in this review. One was from the Philippines, while the others were from the United States and Spain. The sample size for the study in the Philippines was eight, and the data were collected via in-depth online interviews. The study in Spain involved 27 informants, and the data were collected via a focus group discussion. The study from the United States involved ten informants, and data were also collected via a focus group discussion. These studies only interviewed older adults.

Furthermore, most of the data collection was conducted online due to the COVID-19 pandemic. For instance, all the focus group discussions and in-depth interviews for the qualitative studies were conducted online. For the quantitative studies, two studies used the combination of online and mailing methods in the data collection, two studies used a full online method (social media and online-based interventions), one study used only telephone calls, while another utilised the readily available data from a health app. Another study involved paper and pencil interviews; however, whether the data collection was physical or online was not specified.

### 3.2. Measurements of Internet Use and Mental Health

Two studies included social media use in their research [20,24]. The studies mostly touched on the social media status (user or non-user) and the frequency of using social media. For example, respondents were asked whether they used some well-known and common social media platforms in their countries (e.g., number of social media platforms used). Another two studies examined the frequency of communication with family and friends via social technology such as voice and video calls [18,21]. One study examined the frequency and purpose of Internet use [22]. Some common purposes were seeking health information, connecting with others virtually, and managing finances. One study implemented an online exercise intervention for older adults to improve their psychological well-being during the lockdown [19]. Interestingly, there was one study that utilised the passive smartphone communication applications’ data. The number of times that the communication applications were used in each day for the last seven days were the independent variable of the study [23]. 

As shown in Table 3, from seven quantitative studies, loneliness and depressive symptoms appeared to be the most typical mental health outcomes used by researchers. There were four studies that included loneliness [18,20,21,23] and four studies involved depressive symptoms [19,21,22,24] as a mental health outcome. The loneliness scale by de Jong Gierveld and van Tilburg and Centre for Epidemiological Studies Depression Scale were commonly used to measure loneliness and depressive symptoms. However, no study included loneliness and depressive symptoms at the same time. No single item was reported to measure loneliness and depressive symptoms. Next, two studies evaluated mental health in terms of life satisfaction [19,21]. One study assessed satisfaction with a life scale and another study utilised a single-item measure to assess life satisfaction. One study reported flourishing [19], global measure of mental health [19], suicidal ideation [24], and social well-being [23] as the study’s mental health outcomes. 

### 3.3. Thematic Summary

#### 3.3.1. Individual Differences in Mental Health during the COVID-19 Pandemic

The summary from the qualitative studies indicated that the pandemic had caused loneliness and a sense of isolation due to the absence of physical contact [25,26,27]. However, not all experienced loneliness as some, especially those who are digitally literate, might find that conducting activities virtually are enjoyable and satisfying [25,27]. The COVID-19 pandemic also caused the informants to have difficulty in time management, excessive boredom, sadness, and anxiety [27]. Chen and colleagues noticed that the informants had problems of uncertainty, were unprepared for the new normal, and uncertain about the future [26].

Three quantitative studies summarised the individual differences in mental health during the COVID-19 pandemic [20,23,24]. Bonsaksen and colleagues found that the older group (60 and above) had lower social and emotional loneliness levels than the younger age groups during the COVID-19 pandemic [20]. These findings were similar to a study by Wetzel and colleagues when they found that people aged 60 and above had lower levels of loneliness and higher levels of social well-being than other younger age groups [23]. Furthermore, another study also proved that older people aged 65 and above were associated with lower depressive symptoms and suicidal ideation [24]. In terms of gender differences, Bonsaksen and colleagues reported that males had higher social and emotional loneliness [20]. In contrast, another study reported higher levels of loneliness among females [23]. Interestingly, no association was found between gender, depressive symptoms, and suicidal ideation in Hong Kong [24]. 

In terms of socio-economic characteristics, two quantitative studies reported an inverse association between education level and loneliness. Higher levels of education were associated with lower levels of loneliness [20,23]. Furthermore, higher levels of education were associated with higher social well-being [23]. However, in Hong Kong, a higher level of education (college and above) and the highest monthly household income group were associated with higher odds of depressive symptoms [24]. Finally, the studies also concluded that living arrangement and marital status were associated with mental health during the COVID-19 pandemic as cohabitation, partnership, and marriage were associated with higher levels of social well-being and lower levels of loneliness [20,23,24]. 

#### 3.3.2. Patterns of Internet Use

The findings from three qualitative studies were similar when the informants emphasised the importance of ICTs in making them continue to virtually connect with others during the lockdown period [25,26,27]. Social technology applications such as messaging and video calls were part and parcel of their daily life to keep them connected with their family and friends. A quantitative study also showed that the respondents tended to spend most of their time on social media and messaging apps, with approximately an average of 368 min per week spent on communication applications; however, the breakdown of time used by different age groups was not mentioned [23]. Furthermore, the findings from two qualitative studies reported that the Internet was used to seek for health-related information, especially information on COVID-19 [25,26]. Some informants also highlighted their experience using the Internet such as grocery shopping, ordering food, virtually joining religious activities, and learning new recipes online [25,26].

Yang and colleagues discovered that most older adults were not social media users before the pandemic [24]. In terms of time spent on social media by different age groups, Yang and colleagues discovered that older individuals tended to spend less time on social media than younger people [24]. Next, Bonsaksen and colleagues reported that most of the respondents used social media several times a day, and they had an average of four types of different social media applications. However, the findings based on different age groups were not reported [20]. According to Wallinheimo and Evans, email, shopping, making voice and video calls, and managing finances were crucial online purposes among middle-aged and older-aged Internet users. They also discovered that women, those with a high educational level, and retirees were associated with daily Internet use [22].

#### 3.3.3. The Relationship between Internet Use and Mental Health: Comparison by Different Age Groups 

Three quantitative studies compared the relationship between Internet use and mental health between different age groups. First, Wetzel noted that the time used for smartphone communication among the older participants was associated with a higher social well-being and lower loneliness. However, frequent smartphone use in young participants was linked to less social well-being and higher loneliness [23]. Yang and colleagues also discovered similar results when they reported that social media use was inversely linked to depressive symptoms in older adults but not in the younger group [24]. Furthermore, Bonsaksen and colleagues reported that the number of social media platforms used was associated with lower social loneliness among those aged 60 and above; however, it was associated with higher emotional loneliness among respondents aged 18–39 years old [20].

#### 3.3.4. Internet Use and Mental Health

A study showed that older adults could benefit from communication using ICTs during the pandemic, as the results reported that the elderly who rarely or never communicated with others using ICTs had higher levels of social and family loneliness [18]. Two studies examined the relationship between Internet use and mental health in middle age and older adults, but relationships by different age groups were not reported. First, in a sample older than 40 years old, those who used the Internet less frequently to contact friends and relatives reported higher loneliness, lower life satisfaction, and more depressive symptoms [21]. Second, the frequency of Internet use was associated with lower levels of depressive symptoms and higher levels of quality of life. Regarding the purpose of Internet use, a higher frequency of communication purposes was associated with higher levels of quality of life; however, health-related searching and government services information were linked with higher depressive scores [22].

According to a qualitative study, the informants mentioned that they felt a sense of self-satisfaction and higher self-efficacy as they learned to use ICTs. Furthermore, Internet use also promoted higher well-being. Using the Internet to join activities virtually as a way to mitigate the risk of COVID-19 infection, and as a source of distraction and entertainment could help them pass the time easily [27]. Not all studies proved that Internet use effectively promoted well-being, as one intervention study showed that online exercise intervention on older people did not provide any effect on flourishing, life satisfaction, or depressive symptoms [19]. 

#### 3.3.5. Difficulty in Using ICTs

Two studies highlighted the difficulties faced by the respondents while using the Internet. First, a qualitative study reported that older people often have difficulty using a mouse and frequently need assistance from others to connect to the Internet. Additionally, age-related changes such as poor vision and back pain also caused them to have difficulties in using ICTs. Some experienced excessive eye tiredness and back pain after using the computer for a long time [25]. Furthermore, Wallinheimo and colleagues identified several reasons for middle-aged and older adults to not use the Internet such as not being digitally literate, they perceived the Internet as unsafe, and did not have access to good equipment and broadband [22]. 

## 4. Discussion

### 4.1. The Comprehensive Measurement of Internet Use 

The present study identified several important elements that should be included in future studies, which are frequency (or duration) of using the Internet, purpose of using the Internet, frequency (or duration) of each purpose, and number(s) of social media accounts. There are no standardized instruments to measure Internet use in older adults, and therefore, most studies had a different measurement of Internet use. Some of the items were only provided with “yes/no” dichotomous responses, limiting an in-depth understanding of Internet use. For example, one study only asked if the participants were using ten of the social media channels listed without asking them the reasons for not using it if they answered “no”. This is problematic as the granularity and in-depth information of reasons for not using the Internet are unavailable for investigation. Quittschalle and colleagues explored the factors associated with Internet use in older adults. They found that larger social networks, lower depressive symptoms, higher quality of life, and a greater number of chronic illnesses were associated with higher Internet use [28]. In addition, Gell and colleagues identified that impaired physical capacity and disability such as visual impairment and memory problems were associated with lower technology adoption in older adults [29]. 

One important domain of Internet use was not commonly found in the included studies: Internet skills. Internet skills should be included in the Internet use measurement as it is the primary determinant of Internet use in older adults [30]. Studies have concluded that older adults with a higher socioeconomic status (e.g., higher education level and household income) tend to have higher Internet skills [31]. Internet skills among older adults are not as good as those of younger people due to the digital divide and digital inequality. Older adults who were poor, low educated, illiterate, and have difficulty gaining Internet access had extreme difficulties using the Internet [32]. Consequently, older adults have lower self-efficacy and confidence in connecting to the Internet and understanding how the Internet works [33]. The measurement of Internet skills could be as simple as one item of self-reported Internet skill or a validated Internet skill scale consisting of strong theoretical consideration. For example, van Deursen and colleagues proposed four Internet skills that should be evaluated: operational, formal, information, and strategic Internet skills [34]. The multidimensional measurement of Internet use is warranted as different aspects of Internet use might have different relationships on health and well-being. Hunsaker and colleagues suggested the approach of Internet use measurement should include the following aspects: years of use, frequency of use, context of use, Internet skills, and types of Internet use [35]. 

### 4.2. The Possible Positive Effects of Internet Use on Well Being in Older Adults during the COVID-19 Pandemic

Most studies found a positive association between Internet use and mental health in older adults, mainly when older persons used the Internet for social networking purposes. The results were motivating as they provided possible solutions to maintain the well-being of older adults during the COVID-19 pandemic, which is to continue supporting older adults by encouraging them to continuously connect virtually with their friends and family during the lockdown period. Older adults were the ones that were greatly affected during the pandemic as they were the vulnerable group; however, it was a necessity for them, in order to continue having social and emotional support during the lockdown period. A recent study reported that loneliness among older adults increased after the announcement of a lockdown [6]. Women, those living in poverty, living alone, with no children, unsatisfied with neighbour interactions, and physical distancing were associated with higher levels of loneliness in older adults [6].

The relationship between social media use and mental health differs between older and younger adults. Several studies have found that social media use in younger adults did not promote good mental health as it increased the level of loneliness [20,23,24]. The difference could be due to the purpose of social media use in both groups. Older adults often use social media to stimulate social interactions, where they feel contented and happy by connecting with others virtually. However, younger adults often use social media to compensate for their loneliness. They scroll through social media when they feel lonely and hope that they can compensate for the feeling by frequently browsing their social media applications. Unfortunately, most younger people who frequently use social media are associated with poor social skills. Therefore, they are unable to achieve better psychological well-being by frequently checking social media.

### 4.3. Barriers of Older Persons in Adopting ICTs and the Need of Having More Older Adults-Friendly ICTs

Several studies have highlighted the reasons why older people are not keen on using the Internet during the lockdown period. Some common reasons from the included studies were lack of knowledge on Internet use and physical disability. Vaportzis and colleagues identified several barriers of older adults in using digital technology in their qualitative studies: lack of Internet literacy, lack of confidence in coping with the usage of digital technology, poor health status, and non-affordable digital technology [33]. Similar to the previous findings, Yazdani-Darki and colleagues concluded that physical and mental health deterioration, digital technology literacy, limited access to technology, and negative attitudes toward technology were common barriers to using digital technology among older adults [36]. 

Developing more elderly-friendly social technology is possibly one of the solutions to engage more older adults in utilising the Internet for social interactions. According to Chamber and colleagues, several issues need to be considered to make social technology more elderly-friendly such as size of text, letters, and symbols should be big, number of text per page should be short, the contrast between text/graphics and background should be obvious, jargon be avoided in the text, and the size and relevance of images should be appropriate [37]. Furthermore, Yusof and colleagues also recommended that the design framework of elderly-friendly applications should be based on the visual (size of text and symbol, colour contrast), cognitive (user-friendly and less cognitive ability required to operate the application), dexterity (voice recognition application instead of fully operated by fingers or hand), and audio (appropriate audio to make the application more interesting) aspects [38]. 

### 4.4. Directions for Future Research

Future research will benefit whether there is a standardised scale to measure Internet use in older persons. To the best of our knowledge, there is no standardised measurement of Internet use, particularly in older adults. This should be resolved as different measurements from different studies make study comparisons difficult. Next, researchers should sit down and discuss reaching a consensus together on the standardised measurement of Internet use based on a sound theoretical basis. Aside from allowing a reasonable comparison of results across different studies, a standardised scale allows researchers to measure key facts, concepts, and phenomena correctly [39].

Subsequently, Internet skills should be acknowledged as the main factor in determining Internet use in older adults. Future studies should have more action research in improving digital literacy among older adults, particularly in vulnerable older adults such as those living in poverty, rural areas, and living alone, and living with poor health status. Aside from increasing Internet usage in older adults by narrowing the digital gap and promoting digital equity, action research on digital inclusion could also increase the self-efficacy and self-confidence of older adults in using digital technology and make digital technology part and parcel of their life, just like those in the younger age group [40,41].

Undeniably, future studies on the association between Internet use and health status in older adults would be more interesting by utilising passive data. Only one study among the inclusive studies involved passive data analysis. Passive data are mostly collected from smartphone applications. Passive data have several advantages compared to survey data. Passive mobile data collection allows for a higher frequency of data collection, and therefore makes the data even richer. Additionally, passive mobile data collection is more respondent-friendly due to fewer survey questions, and most importantly, passive data collection minimises the recall bias and social desirability bias that could lead to higher data accuracy [42]. The types of data collected via passive mobile data collection are varied: these are geolocation, physical movements, online search behaviour, browser history, applications usage, and call and text message history [42].

It also acknowledges that most older adults are unable to use the Internet due to the difficulty in using electronic gadgets such as smartphones, tablets, or laptops. Future research should also develop elderly-friendly and affordable smartphone applications or electronic devices. Smartphone applications and electronic gadgets that are elderly-friendly could increase the penetration of smartphone and Internet use among older adults. For example, researchers can develop social media platforms that cater to older people’s needs using a simple interface, large fonts, and well-coloured, with a simple registration process and a non-complicated process to post, like, and share content. A prototype can be developed, and a feasibility study can be conducted with the engagement of older persons, led by a research team, to build suitable applications for older adults.

### 4.5. Limitation and Strength of the Study

Several limitations were identified from this review. First, due to the novelty of the COVID-19 virus, all the studies included were of a cross-sectional nature. A cross-sectional study design cannot conclude the causative direction of a relationship. As the COVID-19 pandemic is still worsening with new variants such as Delta and Omicron, longitudinal studies are warranted to confirm its causative relationship. Next, not all studies have evaluated Internet use by purpose; most studies only examined a part of Internet use such as communication via calls or social media. Consequently, the evidence of the relationship between Internet use for other purposes such as information seeking, grocery shopping, and mental health cannot be concluded. Therefore, future studies should include more purposes of Internet use so that the relationship between the different purposes of Internet use and mental health can be identified.

Despite all the limitations mentioned beforehand, this review has several strengths. First, this review is particularly relevant and close to many experiences on a global scale under pandemic restrictions for COVID-19. Thus, the topic is of contemporary relevance and makes a valuable contribution to knowledge at this time. Second, this review included studies from different research designs (i.e., quantitative and qualitative). Therefore, the thematic synthesis could be conducted through multi-faced and comprehensive perspectives. Last but not least, the included studies were recent and peer-reviewed, and therefore, this review has a high academic rigour.

## 5. Conclusions

The relationship between Internet use and mental health in older adults might depend on the purpose of Internet usage. In short, according to the evidence extracted from this review, Internet use for communication purposes seems to be associated with better mental health in older adults during the COVID-19 pandemic. The definition of “communication” in this review refers to imparting or exchanging information by speaking, writing, texting, or other media. This review provides theoretical implications by confirming the engagement theory of ageing, which suggests that keeping older adults active and connected virtually during the lockdown period is required. Furthermore, this review also provides practical implications by suggesting that communication through the Internet would remedy the psychological distress and social loneliness in older people caused by the social restriction imposed by the COVID-19 pandemic. In terms of social implications, this review suggests a need for older people to adopt a new norm: to utilise the Internet for communication purposes during the pandemic period to keep them psychologically healthy, prevent loneliness, and reduce the risk of catching COVID-19. Communication virtually through the Internet is an excellent strategy to maintain their mental health. However, to achieve this, the barriers of older adults in adopting digital technology such as lack of digital literacy, absence of elderly-friendly digital technology, absence of Internet connection, and poor dexterity must be resolved so that digital technology penetration among older adults can be increased. Essentially, this review also suggests several recommendations for future research: (i) a standardised scale to measure Internet use and more studies using passive data are required to push Internet use and mental health studies in older adults to a higher level; (ii) action research to digitally include seniors and design research to develop more elderly-friendly smartphone applications and electronic gadgets that merit scrutiny to ensure more older adults benefit from digital technology, particularly from the Internet; and (iii) measurements of Internet use should include comprehensive domains instead of only communication domains, in order to evaluate the different Internet use purposes with mental health. Finally, policymakers and Internet service providers should encourage and practice net neutrality and a zero-rating plan to ensure digital quality and facilitate older adults in adopting the Internet.

## Figures and Tables

**Figure 1 ijerph-19-03658-f001:**
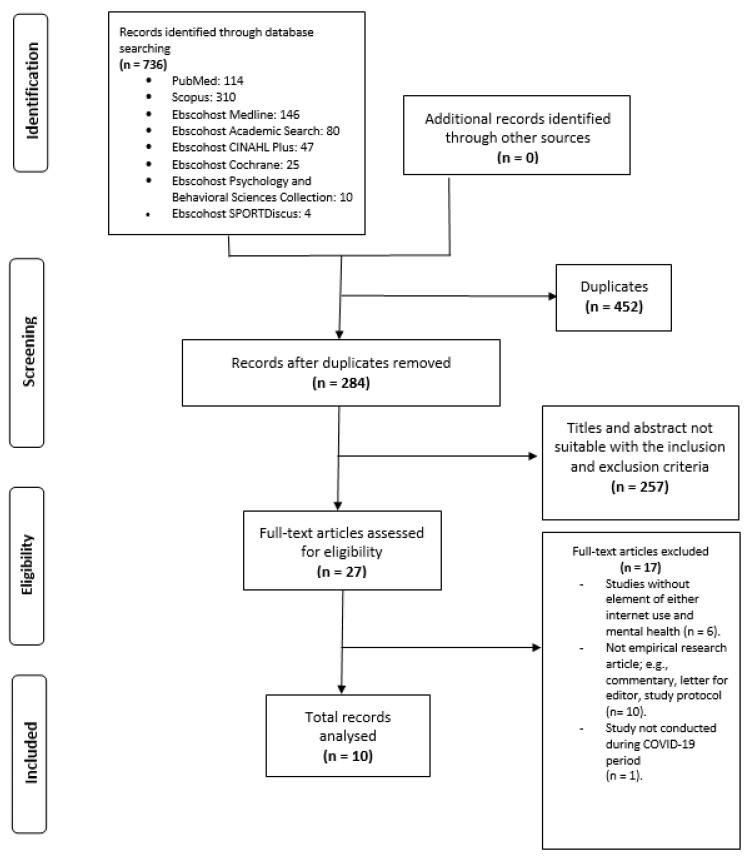
Review flowchart.

**Table 1 ijerph-19-03658-t001:** Search terms.

Variables				Target Group		Time Frame
Internet OR social media OR email OR computer OR smartphone OR social networking	AND	Mental health OR psychological health OR psychological distressed OR psychological well-being OR loneliness OR social isolation	AND	Older adults OR older people OR elderly OR older OR aging OR senior citizen	AND	Coronavirus OR 2019-nCoV OR COVID-19 OR pandemic OR quarantine

**Table 2 ijerph-19-03658-t002:** Inclusion and exclusion criteria.

Domain	Inclusion Criteria	Exclusion Criteria	Rationale
1. Publication year	2019–12 October 2021	Publications prior to year 2019 and after 12 October 2021.	This review only included studies conducted during the COVID-19 pandemic. The first COVID-19 case was reported in year 2019.
2. Publication type	Studies published in peer-reviewed journals only	Studies, reports, or other materials not published in peer-reviewed journals.	To ensure the quality and academic rigour of the included studies.
3. Research design	Both quantitative and qualitative studies were included.	Studies that did not involve any research elements such as commentary or letter for editor.	This scoping review intended to find out to what extent of the literature on Internet use and mental health in older adults during the pandemic period, regardless of research design.
4. Study scope/variables	Studies that involved studying Internet use and mental health. Internet use could be email, smartphone applications, social media, social networking sites, and any online intervention. Mental health covered both positive and negative affective such as quality of life, flourishing, life satisfaction, depressive symptoms, loneliness, and social isolation.	Studies that did not involve Internet use and mental health in a same paper. Studies that did not conduct during COVID-19 pandemic period.	To ensure that the review question is addressed.
5. Target group	Older adults (definition of age following the papers included). Studies that involved general adults were allowed with the condition that older group was also included, or the findings of non-older adults were compared with older adults.	Studies that did not involve older adults at all.	To answer the review question.
6. Location	No restriction	N/A	To answer the review question.

**Table 3 ijerph-19-03658-t003:** Summary of quantitative studies.

Study	Aim	Country/Setting	Design	Sample Size	Sample Characteristics	Internet Use Measure(s)	Mental Health Measure(s)	Principal Findings
1. (Bertić & Telebuh, 2020) [18]	To examine the relationships between ICT communication, social loneliness, and family loneliness in older adults.	Croatia	Cross-sectional	101	- 56.4% females, 43.6% males. - Age range: 66–83 years old. - Mean age: 71.0 ± 4.23. - 63.4% lived with others, 36.6% lived alone. - 41.2% married, 13.1% divorced.	Frequency of using smartphone chat applications (1—never, 2—very rarely, 3—occasionally, 4—constantly).	Loneliness (Social and Emotional Loneliness Scale)	Those who communicated using ICTs “constantly” and “occasionally” reported lower in social and family loneliness than respondents who “never” communicated.
2. (Beau-champ et al., 2021) [19]	To examine if online group-based exercise program relative to an online personal program and waitlist control improved the psychological health among low active older adults.	Canada	Randomised controlled trial	241	- 187 males, 54 females. - Age range: 65-94 years old. - Mean age: 73.0 ± 5.42.	Online delivery of group-based and personal exercise program.	- Flourishing (Diener and colleagues’ 8-item measure).	Flourishing was not improved after the implementation of online-based exercise program.
3. (Bon-saksen et al., 2021) [20]	To examine the relationship between social media use and loneliness within different age groups.	Norway, the United Kingdom, the United States, and Australia	Cross-sectional	3810 (21.9% were older adults age 60 and above)	- 79.6% females, 20.4% males. - 37.2% were under the age of 40, 40.7% were aged 40–59. - 74.0% had education at least Bachelor’s degree. - 70.0% was working full-time or part-time. - 61.4% lived with a spouse or partner.	- Whether respondents had used any of the 10 social media channels (yes/no). - How often they had used social media in general (1—monthly/less frequently, 2—weekly, 3—a few times per week, 4—daily, 5—several times daily).	- Loneliness (The loneliness scale by de Jong Gierveld & van Tilburg)	Higher frequency of social media use was associated with lower social loneliness in older participants, whereas the reverse association was found among the younger participants.
4. (Hajek & König, 2021) [21]	To examine the relationships between frequency of contact with friends and relatives via Internet, loneliness, life satisfaction, and depressive symptoms among middle-aged and older adults.	Germany	Cross-sectional (secondary data analysis)	3134 (Percentage of older respondents was not specified)	- 49.4% females, 50.6% males. - Mean age: 67.6 ± 9.7. - Age range: 46–98.	The frequency of contact with friends and relatives via Internet (never; 1 to 3 times a month; once a week; less often; several times a week; daily)	- Loneliness (The loneliness scale by de Jong Gierveld & van Tilburg). - Life satisfaction (SWLS). - Depressive symptoms (CES-D).	Less frequency of contacts was associated with higher loneliness, lower life satisfaction, and higher depressive symptoms.
5. (Wallin-heimo & Evans, 2021) [22]	To examine the associations between frequency and purpose of Internet use, depressive symptoms, and QoL in middle-aged and older adults.	England	Cross-sectional (secondary data analysis)	3491 (Percentage of older respondents was not specify)	- 57% females, 43% males. - Mean age: 67.2 ± 5.23.	- Frequency of Internet use (more than once a day, once a day, once a week, and less than once a week) - Purpose of Internet use (e.g., information searching and communication).	- Depressive symptoms (CES-D-SF). - Quality of Life (CASP-12)	- More frequent use of the Internet was associated with higher QoL. - Using the Internet for communication purposes was linked with higher QoL, While information searching was linked to lower QoL and higher depression symptoms.
6. (Wetzel et al., 2021) [23]	To examine the associations between smartphone communication, self-reported loneliness and social well-being across different age group.	Germany	Cross-sectional	364 (11.8% were older adults age 60 and above)	- 52.5% females, 46.2% males. - Mean age: 42.6 ± 13.17. - 67.0% with high educational level.	Passive smartphone communication app data.	- Loneliness (Loneliness Scale-SOEP) - Social wellbeing (WHOQoL-Brief)	Higher social media use time was associated with higher loneliness and lower social well-being in younger participants, whereas the opposite association was found for older adults.
7. (Yang et al., 2021) [24]	To examine the relationships between social media use, depressive symptoms and suicidal ideation among adults.	Hong Kong	Cross-sectional	1070 (63.9% were older adults age 56 and above)	- 67.7% females, 32.3% males. - 69.9% cohabiting or married.	Hours spent per day on social media.	- Social loneliness (The loneliness scale by de Jong Gierveld & van Tilburg)- Depressive symptoms (CES-D-10)	- Negative association was found between social media use and depressive symptoms in older people, but not in younger people. - There was no association between social media use and suicidal ideation in both younger and older groups.

Note: ICT; information and communication technology, SWLS; satisfaction with life scale, CES-D; Centre for Epidemiologic Studies Depression Scale, CES-D-SF; Centre for Epidemiologic Studies Depression Scale—Short Form; CASP; Control, Autonomy, Self-realization, and Pleasure scale, QoL; quality of life, SOEP; The German Socio-Economic Panel.

**Table 4 ijerph-19-03658-t004:** Summary of qualitative studies.

Study	Aim	Country	Data Collection Method	Sample Size	Sample Characteristics	Internet Use	Analytical Approach	Mental Health-Related Themes Reported
1. (Castillo et al., 2021) [25]	To describe the experience of older adults who were living alone in using social media technologies.	Philippines	In-depth online interview	8	- Age range: 62–71 years old. - 6 females, 2 males.	Social media platform usage	Iterative processes distinct in the hermeneutic circle adopted in van Manen’s (2014) approach.	- Gratitude in using social media (pleasure and joy from the use of social media). -Choosing battles (having difficulties and limitations in using ICTs) - Managing connections (connecting with others using social media made them alone but not lonely; obtained emotional support). - Belongingness (continue to maintain relationship with family members).
2. (Chen et al., 2021) [26]	To explore how pre-frailty and frailty older adults address the challenges of COVID-19 pandemic.	United States	Online focus group discussion	10	- Age range: 66–84 years old. 3 males, 7 females.	Information and technology use	General inductive analytic method.	- Social isolation was mitigated by the use of social technology.
3. (Llorente-Barroso et al., 2021) [27]	To describe the impact of the ICT use on the emotional well-being of elderly people during lock down.	Spain	Focus group discussion	27	- Age range: 60–77 years old. - Nine men, 18 women.	The use of ICTs	Thematic analysis	- Emotional support obtained while connecting with others via ICTs. - Using ICTs helped to reduce the anxiety related to COVID-19 infection. - ICTs as an entertainment tool to distract users during the lock down. - Self-satisfaction obtained while learning how to use ICTs - Use of ICTs in pursuing personal hobbies and concerns.

Note: ICT; information and communication technology.

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
