# Peer review of "For Better or for Worse? A Scoping Review of the Relationship between Internet Use and Mental Health in Older Adults during the COVID-19 Pandemic"

_ijerph, 2022, doi:10.3390/ijerph19063658_

Round 1
Reviewer 1 Report
ARTICLE: For Better or For Worse? A Scoping Review of The Relationship Between Internet Use and Mental Health in Older Adults During The COVID-19 Pandemic AUTHORS:Dear author,
First of all, congratulations on the work done and the effort involved in conducting research of this kind.
It provides good conceptualisation, design, good analysis and discussion along with limitations,
Congratulations on the work done.
However, I would suggest a number of recommendations:
-Incorporate what theoretical implications this work has for the scientific community.
-Incorporate what practical implications this work has for the community/society/population.
-Incorporate the social implications of this work.
-Incorporate strengths of your work.
Reviewer 2 Report
Comments and suggestions for authors
I read with interest the manuscript entitled ‘For Better or For Worse? A Scoping Review of The Relationship Between Internet Use and Mental Health in Older Adults During The COVID-19 Pandemic’. The authors conducted a scoping review of studies on the use of the Internet by the elderly during the Covid-19 pandemic and how it influenced their mental health, especially in conditions of social isolation.
Introduction
In part 1.1 the authors should improve the references on the impact of the Covid-19 pandemic on the older adults and cite more studies in this regard.
In part 1.2 authors should mention the authors (the research) according to which ‘Internet usage has risen from 65 40% to 100% compared to the pre-lockdown period’ (lines 65-66).
Methodology
Is clearly and completely presented, the authors mentioned and described the steps of a scoping review.
Results
The design and the findings of each research included in this scoping review are specified.
I appreciate that the results are presented in detail and systematized, and are easy to follow.
The authors should formulate the sentence from lines 222-223 more clearly. As I understand, there was no article which measured both loneliness and depressive symptoms (at the same time). The authors should clarify this.
Discussions
Are presented in a systematized manner and are supported by the articles included in the current scoping review.
The authors identify some limitations of the respective studies and also outline some directions for future research and practical applications.
Line 382 – the authors should mention the authors of ‘a recent study’
References
The references regarding the topic included in the scoping review are very current.
The references should be written (in the text of the manuscript and in the list of references) according to the ‘Instructions for Authors’ section.
Date of review: 5.03.2022
Reviewer 3 Report
First of all, I wish to congratulate the authors for conducting such a pertinent and important study in a timely fashion. A more nuanced understanding of the intersection of older people’s Internet use and their mental health is of pivotal importance, particularly in light of critical social issues like the necessity of COVID-19 safety measures, the loneliness epidemic, and the raging infodemics. Overall, this study is of very high impact and could help enrich the literature and inform better policymaking. Please find my comments below and address them properly, as I believe that, while they are minor in nature, shedding light on these concerns could help the authors further strengthen their work and the readers better appreciate the study.
What does “Internet use” entail? Please offer a detailed definition of the key terms studied.
A related concern centers on the search terms adopted in Table 1. It seems that terms like “tablets”, “applications or mobile applications”, “smartwatches”, etc. were not considered in the search? The same concern applies to other themes studied (e.g., mental health vs. depression/anxiety/distress, etc.). What is the rationale behind the choice of terms adopted? Please elaborate on why some terms were used while others were not.
The authors mentioned, “We developed two templates to populate data from the included studies” (Line 148). How were these templates developed and what validation processes were adopted?
In terms of results, the authors state that “Our review has revealed Internet use for communication purposes seems associated with better mental health in older adults during the COVID-19 pandemic.” Please provide a clear definition/conceptualization for concepts like “communication purposes”.
Some minor issues. Context, e.g., the characteristics of the older people studied, should be better described in statements like “…more than half of the older adults living with morbidity had moderate to severe levels of loneliness” (Line 57-58).
Reviewer 4 Report
Thank you for your paper - the topic is particularly relevant and close to the experiences of many on a global scale under pandemic restrictions for Covid-19. Thus the topic is of contemporary relevance and makes a valuable contribution to knowledge at this time.
Also by reviewing and drawing together such contemporarily specific sources in this way, a resource for further exploration of this issue is very well provided.
The abstract is clear and useful. The introduction is also appropriate to the study.
The claim at lines 30-32 that "Coronavirus Disease 2019 (COVID-19) has impacted the world since December 2019. Due to the absence of effective medicines and a slow vaccination rate during the initial occurrence of the pandemic," is limited as different global locations had different rates of vaccination uptake - for example the UK had a fast approval and uptake rate of vaccination. With this in mind I recommend changing this to either reflect that there was "an internationally variable rate in the uptake of....." and/or explicitly state the geographic location you are specifically referring to with this claim.
Line 78 - reference issue - " (S. S. M. Lam et al., 2020)" this doesnt fit with the consistency of the in text citation style adopted throughout. Not sure what the "S.S.M" bit is for? please revisit this and check all in text citations and references.
The diagrams and tables are useful and informative though these could be moved to appendices or supplementary files if space becomes an issue.
The pandemic specificity as stated offers a novel value to the piece.
The methodology is sound - though there is no reference to ethical approval process - even as a cat 0 review project i would normally expect to see an approval reference? Please check and address whether this is available / needed in this case.
The discussion is good and valuable and largely identifies the value and considerations within the piece.
This claim at line 410-411 is an overstatement and not grounded- there are other ways but it does hold strong potential - recommend tha tyou reword to refelct thi s "Developing more elderly-friendly social technology is the only solution to engage more older adults in utilising the Internet for social interactions. "
The conclusions are clear and make sense.
Overall i believe that this article merits publication in the journal, following the identified amendments being considered and made. Thank you for your piece.
